



**DETECTION AND ATTRIBUTION OF FLOOD TRENDS IN MEDITERRANEAN**
**BASSINS**
Tramblay, Yves[1]
Mimeau, Louise [1]
Neppel, Luc [1]
Vinet, Freddy[3]
Sauquet, Eric [2]
[1] HSM (Univ. Montpellier, CNRS, IRD), 300 Av. du Professeur Emile Jeanbrau, 34090,
Montpellier, France
[2] IRSTEA, UR RiverLy, Centre de Lyon-Villeurbanne, 5 rue de la Doua CS 20244, 69625
Villeurbanne, France
[3] GRED (Univ. Paul Valéry, IRD), 2 rue du Pr Henri Serres, 34000 Montpellier, France



**Abstract**
Floods have strong impacts in the Mediterranean region and there is a questioning about a
possible increase in their intensity due to climate change. In this study, a large database of 171
basins located in South France with daily discharge data with a median record length of 45 years
is considered to analyze flood trends and their drivers. In addition to discharge data, outputs of
precipitation, temperature, evapotranspiration from the SAFRAN reanalysis and soil moisture
computed with the ISBA land surface model are also analyzed. The evolution of land cover in
these basins is analyzed using the CORINE database. The trends in floods above the 95[th] and 99[th]
percentiles are detected by the Mann-Kendall test and quantile regression techniques. The results
show that despite the increase in extreme precipitation reported by previous studies, there is no
general tendency towards more severe floods. Only for a few basins, the intensity of the most
extreme floods is showing significant upward trends. On the contrary, most trends are towards
fewer annual flood occurrences above both the 95[th] and 99[th] percentiles for the majority of basins.
The decrease in soil moisture seems to be an important driver for these trends, since in most
basins increased temperature and evapotranspiration associated with a precipitation decreases are
leading to a reduction of soil moisture. These results implies that the observed increase in the
vulnerability to these flood events in the last decades is mostly caused by human factors such as
increased urbanization and population growth rather than climatic factors.








**Keywords:**
**Floods, trends, France, Mediterranean, soil moisture**




## 1. INTRODUCTION

A number of studies have now established that extreme precipitation could increase due to climate change in particular in the Mediterranean (Westra et al., 2013, Polade et al., 2017, Ribes et al., 2018, Tramblay and Somot, 2018). Changes in extreme rainfall would be caused by an increase in the precipitable water content in the atmosphere, related to increasing temperatures, according to the principle of Claussius-Clapeyron thermodynamics (Drobinki et al., 2016, Pfahl et al., 2017). Nevertheless, this relationship has a high variability in space, related to temperatures and available humidity (Prein et al., 2016). Several studies observed an increase in the number of dry days associated with increased rainfall intensities, suggesting that dry periods in these areas would become longer, but that precipitation could be more extreme when they occur (Paxian et al., 2015, Polade et al., 2017). Nevertheless, the increase in extreme rainfall would not offset the decrease in precipitation totals, as the drop in cumulative rainfall associated with the decrease in the frequency of low to moderate rainfall is expected to predominate over the gains resulting from the intensification of extreme precipitation (Polade et al., 2017).

Beside changes in precipitation, an increase in rainfall intensity does not necessarily imply an increase in flood risk (Ivancic and Shaw, 2015, Woldemeskel and Sharma, 2016). Indeed, for a given rainfall accumulation, the runoff coefficient can be very variable in time and space in different basins due to complex interactions between precipitation and infiltration processes on hillslopes which can strongly modulate flood magnitude (Woldemeskel and Sharma, 2016, Wasko and Sharma, 2017, Bennett et al., 2018). Most global studies on flood trend indicate a decrease in flood intensity (Do et al. 2017, Wasko and Sharma, 2017, Sharma et al., 2018). Yet, these trends are highly variable in space for different regions of the globe (Yin et al., 2018, Najibi and Devineni, 2018). The attribution of these trends is rather uncertain, while Yin et al. (2018) relate an increase in floods with increased temperatures; Najibi and Devineni (2018) or Hodgkins et al. (2018) conclude that trends in the flood frequency and duration can be mostly attributed to long-term climate variability. Nonetheless, as noted by Whitfield (2012), flood generating processes do not take place at the global but rather a relatively local scale, making generalizations about flooding in future climates difficult and uncertain. For Mediterranean basins, Blöschl et al. (2017) indicate later winter floods and Mangini et al. (2018) noted a tendency towards increasing



flood magnitude and decreasing flood frequency. These finding are consistent with trends
detected by Mediero et al. (2014) in Spain and Giuntoli et al. (2012) for the South of France.

While much work has been done to estimate future climatic conditions, it is not clear about
possible changes in hydrological variables including surface conditions that can strongly
modulate climatic trends (Knighton et al., 2017). In particular, it is known that in many
catchments the initial soil moisture conditions prior to flood events play a key role in flood
generation (Brocca et al., 2008, Tramblay et al., 2010, Raynaud et al., 2015, Woldemeskel and
Sharma, 2016, Wasko and Sharma, 2017, Uber et al., 2018, Wasko and Nathan, 2019) and its
temporal change has not been much analyzed up to now. Between two episodes of rain, the base
flow of the perennial rivers originates from the draining of the water contained in the soils and for
some basins from the aquifers. The capacity of the soil to contain water and restore it to generate
runoff depends on its characteristics (texture, structure, porosity ...) but also on the amount of
water it already contains at the beginning of a rain episode. Thus, a quasi-saturated soil will not
be able to store a lot of water, which, being unable to infiltrate, will contribute directly to runoff.
In most cases, there is a non-linear relationship between the flow rate and the initial saturation
state of the soil, usually with a threshold value of moisture above which a rapid flow response to
a rainy episode is observed (Norbiato et al., 2008, Viglione et al., 2009, Penna et al., 2011).
Difference in soil types could induce different relationships between floods and initial conditions
(Grillakis et al., 2016, Camarasa-Belmonte, 2016). For intermittent (seasonal runoff only) and
ephemeral streams (runoff only after a rain event), the impact of antecedent soil moisture is more
complex and strongly dependent on the soil type and geological context (in the presence of karst
in particular). In smaller basins, the impact of initial soil moisture content is usually not
significant and it increases with catchment size (Zhang et al., 2011).

For some Mediterranean basins, the increase in heavy rainfall associated with a reduced number
of rainy days could decrease the soil water content and therefore increase infiltration capacity,
hence reducing runoff. On the other hand, more intense rains in urbanized, impervious areas or
on bare soils that are subject to crusting effects could increase runoff and therefore the magnitude
of floods. It is therefore necessary to use hydrological or surface models capable of representing
these processes. Quintana-Seguí et al. (2011) using the ISBA land surface scheme with different



downscaling methods found a future increase in floods corresponding to a 10-year return level in
southern French basins, but with different magnitudes depending on the basins. Camici et al.
(2017), in a study on the impacts of climate change on floods in central Italy, noted a greater
sensitivity of basins with permeable soils to changing climatic conditions. Similarly, Piras et al.
(2017) in Sardinia found that impermeable and flat sub-basins are predicted to experience more
intense flood events in future scenarios, while more permeable and steep sub-catchments will
have an opposite tendency. However, there are systematic differences between projections of
changes in flood hazard in south Europe (Italy, Greece, Iberian Peninsula) in most European and
global studies using large-scale hydrological models (Kundzewicz et al., 2017). Indeed some
studies points towards an increase in southern Europe (Quintana-Seguí et al., 2011, Alfieri et al.,
2015) while others suggests a decrease (Donnelly et al., 2017, Thober et al., 2018). This is due to
different GCM, RCM, scenarios and downscaling approaches but also the use of large scale
hydrological model usually not calibrated and validated for all basins. This type of global (or
large scale) hydrological model (LISFLOOD, VIC, HYPE…) is usually not adapted to small
river basins less than 500 km², which is the typical catchment size found in the Mediterranean
region.

Prior to make future projections on flood hazard, there is a need to understand the main drivers of
changes for floods and the links between floods and climate characteristics (Merz et al., 2014).
Indeed, understanding the potential flood drivers and their changes may be more relevant than
predictions of uncertain flood changes as noted by Blöschl et al. (2016). The objective of this
study is to analyze trends in floods characteristics for a large sample of French Mediterranean
basins and to relate these trends to climate and land use dynamics. This is done using statistical
tests for the detection of trends and quantile regression models to relate high discharge quantiles
to different climatic drivers.

**2. DATA**

171 basins located in south France are selected with a minimum of 20 years of daily discharge
data. The selection of basins is based on the availability of long time series of daily discharge and
the selected basins have no significant human influence on flow, from a previous database





elaborated from Sauquet and Catalogne (2011) and Snelder et al. (2013). The median record
length is 45 years and 56 stations have more than 50 years of data, more than 100 stations have
complete years, with less than 5% missing data, between 1970 and 2010. All the catchments
selected have a Mediterranean climate, with a precipitation deficit during summer when the low
flows are recorded. These basins are experiencing flash flood events caused by intense rainfall
events, corresponding to the only region in France when rainfall can exceed 200 mm/day
(http://pluiesextremes.meteo.fr) with the maximum occurrence between September and
November. Most basins have a catchment area lower than 500 km² and located below 1000 m.
(figure 1). The proportion of karstic areas for each basin has been obtained from the BDLISA
database (available here: https://bdlisa.eaufrance.fr/) which provides a delineation of karst
systems in France (Schomburgk et al., 2016). Very common geological formation in the French
Mediterranean region, about 50 gauged basins have more than 50% of their catchment areas with
carbonaceous superficial formations, indicative of Karstic areas. This means that the rainfall-
runoff relationship in this type of basin can be strongly modulated by the presence of karst
(Jourde et al., 2007).

In addition to daily discharge data, different climatic variables have been retrieved from the
SAFRAN reanalysis over France (Quintana-Seguí et al., 2008). This reanalysis based on
observed station data provides rainfall, snowfall, temperature, actual and reference
evapotranspiration for a 8x8km grid over France at the daily time step from 1958 until present.
The SAFRAN reanalysis is used to force the ISBA land surface scheme of Météo-France (Habets
et al., 20008), to provide among other variables the surface and root zone soil moisture at the
same spatial and temporal resolution than ISBA. Tramblay et al. (2010) have shown that the soil
moisture from the root zone simulated by ISBA is an appropriate indicator of soil moisture prior
to flood events in French Mediterranean catchments. The catchment boundaries of the 171 basins
selected have been extracted from the HydroSheds database (https://hydrosheds.org/) providing
flow accumulation and flow direction maps at the 15 arc-second resolution. Then the total
precipitation, rainfall, air temperature, actual and reference evapotranspiration from SAFRAN
and the surface and root zone soil moisture from ISBA have been extracted and averaged over
every catchment.



The evolution of landcover between 1990 and 2018 in the 171 basins was analyzed using the
Corine Landcover inventory (CLC1990 and CLC 2018). Corine Landcover provides an inventory
of 44 classes over the European region (Büttner et al., 2002). CLC1990 and CLC2018 are
respectively based on Landsat-5 (50m spatial resolution) and Sentinel-2 (10m spatial resolution)
satellite images. A limitation of the CLC inventory lies in the difference of accuracy between the
CLC1990 and CLC2018 products, which may introduce an uncertainty in the estimation of the
evolution of the land cover in the studied basins.

**3. METHODS**

Two approaches are considered to evaluate trends. The first approach, presented in section 3.1
thereafter, relies on the Mann-Kendall test applied to the annual number of flood events above
two different percentiles, the $95^{th}$ and the $99^{th}$ computed on the whole time series and also on the
magnitude of these events. Using two different thresholds, which are commonly used for the
analysis of floods, allows considering separately the trends on moderate (above the $95^{th}$
percentile) and more severe (above the $99^{th}$ percentile) flood events.

The second approach presented in section 3.2, is based on quantile regression to estimate the
temporal trend magnitude in the $95^{th}$ and $99^{th}$ percentiles of daily runoff in all stations. The
quantile regression method is also used to relate the change in runoff quantiles to changes in
climate characteristics, hence providing a way to attribute the observed changes to their potential
drivers.

Hydrological years are considered, starting September $1^{st}$ and ending August, 31 of the next
calendar year. Years with more than 5% missing days are removed. For the first approach based
on event characteristics, a de-clustering is required to not include in the flood sample consecutive
daily threshold exceedances that belong to the same flood event. A minimum of 2 days between
two flood events is selected since it is the average duration of rainstorm in the region (Tramblay
et al. 2013). This means, if for two consecutive days the runoff is exceeding the threshold, only
the maximum value is retained. Moreover, different values between 1 and 5 days to separate the
events have been tested but preliminary tests indicated that it did not change the trend results.






### 3.1 Test for trends and regional significance


The Mann–Kendall (MK) test (Mann 1945) is used for the trend detection. Several studies have
noted that the presence of serial correlation may affect the results of trend analysis by increasing
the variance of the test statistic (Khaliq et al., 2009, Renard et al., 2008). To overcome this
limitation, Hamed and Rao (1998) proposed a corrected MK test statistic considering an effective
sample size that reflects the effect of serial correlation. This correction was applied in the present
study. In addition to the MK test, the method of Sen (1968) is considered to estimate the
magnitude of trends. In the present study, trends are considered significant at the 10% level;
however, sensitivity tests performed for $p \leq 0.05$, $p \leq 0.01$ revealed very similar spatial trend
patterns.

The significance level α for a statistical test is related to a single test and is no longer valid when
multiple tests are conducted (Wilks 2016). When the number of tests being conducted increases,
more significant values will be found. The goal of the false discovery rate (FDR) procedure
introduced by Benjamini and Hochberg (1995) is to identify a set of at-site significant tests by
controlling the expected proportion of falsely rejected null hypotheses that are actually true.
Renard et al. (2008), Khaliq et al. (2009) or Wilks (2016) demonstrated that the original FDR is
robust to cross correlations between locations and can work with any statistical test for which one
can generate a p-value. This method is applied to the MK test results to check if the trends are
regionally significant.

### 3.2 Quantile regression


As a complementary approach to detect trends in quantiles but also to investigate the relationship
between floods and explanatory covariates, the quantile regression (Koenker and Basset, 1978)
method is applied. Quantile regression could be seen as the extension of the ordinary least square
(OLS) regression (Koenker and Machado, 1999, Villarini and Slater 2017). In OLS, the
conditional mean of the response variable is modeled with respect to one or more predictors and
the sum of squared errors is minimized. For quantile regression, a conditional quantile of the





response variable is modelled as function of predictor(s), an asymmetrically weighted sum of
absolute errors is minimized to estimate the slope and intercept terms. In the present work, only
linear relationships are considered with one single covariate at a time, while more complex forms
of dependences could also be considered in quantile regression. The approach has been
previously used to detect trends in extreme precipitation or floods by Villarini and Slater (2017),
Yin et al. (2018) or Wasko and Nathan (2019).

Koenker and Machado (1999) introduced the $R^1$ goodness of fit measure for quantile regression
models. As for the $R^2$ in the case of OLS, $R^1$ lies between 0 and 1. Unlike $R^2$, which measures the
relative success of two models for the conditional mean function in terms of residual variance, $R^1$
measures the relative success of the corresponding quantile regression models for a specific
quantile, by comparison with a restricted model (with slope = 0), in terms of a weighted sum of
absolute residuals (see Koenker and Machado, 1999). Consequently, $R^1$ constitutes only a local
measure of goodness-of-fit for a particular quantile rather than a global measure over the entire
conditional distribution, like $R^2$. This measure can help to discriminate between different models
using different covariates (ex: precipitation or temperature). Higher $R^1$ values indicate that the
model fits better to observations. In this study, this criterion is used to identify the best covariates
that could explain the temporal variations in high runoff quantiles.

**4. RESULTS**

**4.1 Climatic and land cover trends**

The climate trends have been analyzed on the whole period of available SAFRAN records,
between 1958 and 2018. From figure 2, It can be seen a significant decrease of annual rainfall in
56 basins, on average of -20%, accompanied by an increase of the frequency in dry days (with
precipitation below 1 mm) for 46 basins. The snowfall is also decreasing in the same proportions
(no shown). The sole exception where an increase in rainfall is found is for the Asse River at
Beyne-Chabrières on the western foothills of the Alps. This station has long time series spanning
from 1983 to 2009, where a +15% trend in annual rainfall is detected over the whole record. Yet,
the detection of this trend might be an artefact since there are several consecutive wet years



between 1992 and 2000. This trend in rainfall can be also seen for the soil moisture trends.
Associated with the precipitation decrease, positive temperature trends are observed for almost all
basins, with an average increase of +0.5°C during the time period 1958-2015. Consequently,
widespread increasing trends in reference and actual evapotranspiration rates over all basins are
observed, similarly as in Vicente-Serrano et al. (2014) in Spain or Rivoire et al. (submitted) for
the whole Mediterranean region. The combined decrease in precipitation with increased
evapotranspiration yields to a decrease in soil moisture for the surface and the root zone layers.
This is in accordance with previous studies over South France such as Vidal et al. (2012) or
Dayon et al. (2018).

About land cover (figure 3), most basins have low urban areas (below 10%) and the basins with
the highest coverage are found mostly in the South East. An increase of urban areas up to +20%
of total catchment surface can be seen between 1990 and 2018 for basins mostly located close to
the Mediterranean coast and in particular in the Provence-Alpes-Côte-d'Azur region. The class
representing discontinuous urban fabric represents 73% of artificialized areas and increased by
+36% between 1990 and 2018. The increase of urbanized areas could have a strong impact on
runoff generation, in particular for small basins, with the increase of impervious surfaces favoring
surface runoff. In contrast, the agricultural and forest land cover can reach 100% of the basin
surface, in particular in the western Tarn regions for agriculture. We can notice a reduction of
forest cover in the Northern Cévennes areas associated with an increase in agricultural surfaces.
When looking in details from the original classification, for some catchments of size 500 km² or
less, the percentage of vineyards could exceed 70% of the total catchment areas in particular for
basins located in the Occitanie region. For almost all basins, the percentage of vineyards has
decreased between 1990 and 2018. The other dominant land use classes related to agriculture are
pastures (27.8% of all catchments), complex cultivation patterns (21.9%) and land principally
occupied by agriculture with significant areas of natural vegetation (27.7%). Forested areas are
mostly represented by broad-leaved forest (35%), coniferous forest (19%) and mixed forest
(14.4%) classes. It must be noted that the land cover change analysis is hampered by the short
duration of the land use maps available, 28 years between 1990 and 2018, and possibly different
sensors during this period leading the different attribution to some land use classes.



## 4.2 Flood trends

To analyze flood trends, all flood events above the 95[th] or 99[th] percentiles of daily runoff computed on the whole time series are extracted. The trend MK test is applied to the number of annual exceedances above these two thresholds and also on the magnitude of the threshold exceedances. From figure 4 it can be seen a general tendency towards a decrease in the annual number of flood events above the 95th percentile, that is significant in 67 catchments, and to a lesser extend also in the number of events above the 99th percentile in 45 catchments. These trends are regionally significant according to the FDR procedure and particularly over the northern ridge of the Cévennes mountainous areas. According to the Sen Slope method to estimate the decrease in the annual number of events above the 95[th] percentile; for most basins the trends are ranging between -0.5 and -1 event per decade. For the most extreme cases the trends can reach up to -2.5 events per decade. Since for all catchments the number of events above the 95[th] percentile per year is 4.5 on average (min =2, max =6, after de-clustering), the magnitude of these trends can be considered low. For the 99[th] percentile the magnitude of trends are similar, with a maximum decrease of -1.4 events per decade, and for most stations on average -0.4 events per decade (with an average annual number of 1.6 events above the 99[th] percentile, after de-clustering). In addition to the trends in the annual number of events, there is also a weak signal of an increase of the magnitude of floods, in particular above the 99th percentile for 16 stations, yet these trends are not regionally significant.

Beside this event-based analysis, the temporal trends in the 95[th] and 99[th] percentiles of the daily runoff time series have been investigated using quantile regression. The approach is complementary but different to the testing of trends on the annual occurrence and the magnitude of the events, since quantile regression allows evaluating the possible changes on the quantiles of daily runoff time series. This analysis reveals that for a majority of catchments, a decreasing trend in these two percentiles is detected. The procedure is to apply a quantile regression of the percentile of interest with time as a covariate, and to validate if the slope of the quantile regression model is significantly different than zero at the 10% level a bootstrap resampling approach (Efron, 1979) has been considered. For the 95[th] percentile, a decreasing trend in 147 stations is found and an increase in only 12 stations. For the 99[th] percentile, 89 negative trends





are found and 15 stations with increasing trends. The relative changes in the 95th and 99th
percentiles are ranging for most stations between 0 and -0.5 as shown on figure 5. The number of
detected trends with quantile regression for the 95th and 99th percentiles is larger than the number
of trends detected with the MK test. However, for many basins the trends in the 95th and 99th
percentiles are of small magnitude and only for the largest trends the MK test also detect
significant changes in the annual number of events above these thresholds.

In an attempt to relate the detected trends to catchment characteristics, the Student t-test has been
used to compare the catchment descriptors between the group of basins with or without trends.
The catchments where decreasing trends in flood occurrence are detected tend to be are larger
catchments (mean size of 369 km² vs. 253 km² for the catchments with no significant trends),
with a lower proportion of karstic areas (33% vs. 41%) and urban areas (1.7% vs 3.79%). Also
more decreasing trends are detected in agricultural catchments than in forested areas. Yet, no
clear link can be found between land cover changes and flood trends, probably due to the short
duration of the land cover dataset available. The only exception is about trends in urbanization,
with a lower increase in urbanization (+0.77% average increase in urban areas) in catchments
where floods are decreasing by comparison with catchments with no flood trends (+1.41%
average increase in urban areas). It must be noted that there is a strong spatial variability of the
observed trends highlighting the complex interplays between the different catchment
characteristics, as similarly noted by Snelder et al. (2013) over France. For example, the
magnitude of the detected trends is not correlated with the different catchment properties. This
implies that it would be very challenging to propose a typology of basins with similar changes in
floods according to catchment properties.

**4.3 Changes in event precipitation and antecedent soil moisture conditions**

For each event, the cumulative catchment precipitation average is computed as the sum of non-
zero consecutive rainy days, on a time window up to 10 days prior to the flood event. The
antecedent soil moisture is taken as the root zone soil moisture corresponding to the day prior the
start of the rainfall event. Figure 6 show the Mann-Kendall test results for these two indicators for
floods above the 95th and the 99th percentiles. An increase of precipitation associated with floods





using both thresholds is observed (for 34 catchments for the 95[th] percentile and 36 catchments for
the 99[th] percentile), associated with a decrease in antecedent soil moisture conditions prior to
floods in up to 40 catchments for floods above the 95[th] percentile. There is a correlation between
the reduction of antecedent soil moisture prior to flood events and the decrease of the annual
number of flood events above the 95[th] percentile (r=0.44), also to a lesser extent for the number
of floods above the 99[th] percentile (r=0.34). Consequently, as observed in Australia by Wasko
and Nathan (2019) it can be hypothesized that the decrease of antecedent soil moisture is an
important driver leading to the reduction of the annual number of floods, despite the increase in
event precipitation already pointed out by several studies in this region (Tramblay et al., 2013,
Ribes et al., 2018, Blanchet et al., 2018). Indeed, for 12 catchments an increase of event rainfall
is detected when for the same catchments a decrease in the annual number of events above the
95[th] percentile is also observed. It is also the case of 11 catchments for the events above the 99[th]
percentile with an increase of event rainfall accompanied by a decrease in the annual number of
events. However as shown before, the increased event precipitation for several basins is probably
the cause of higher flood magnitudes for the most severe events (above the 99[th] percentile).

**4.3 Explanatory covariates for high runoff quantiles**

To test the influence of different covariates on the variation of the 95[th] and 99[th] percentile values,
quantile regression models using time, temperature, soil moisture from the root zone, actual
evapotranspiration (AE), reference evapotranspiration (ET0) and precipitation have been
compared. The goal here is not to select the best covariates for each station but to identify
relevant covariates at the regional scale. Since climatic covariates could influence the
hydrological response at different time scales (Mediero et al., 2014, Villarini and Slater 2018,
Wasko and Nathan, 2019), three different aggregation periods to compute moving averages have
been compared. For the event scale, the different covariates have been averaged with a 3-day
time lag preceding each event. At the monthly time scale representing the seasonal variability, the
covariates have been averaged in the same manner but on 30 days. Finally, for the annual time
scale the covariates have been averaged over 365 days. At the event scale, the precipitation rather
represents the intensity of rainfall during the event than the preceding soil moisture. On the
opposite, for the monthly and annual aggregation periods the precipitation is here a proxy for soil





moisture and its long term variability. To test which covariate provides the best reproduction of
the observed 95$^{th}$ and 99$^{th}$ percentiles of the daily discharge time series, the $R^I$ metric is
computed, for each covariate, between the quantile regression model built with the covariate and
a constrained model with a constant slope (0).

The results are plotted on figure 9. A similar pattern can be seen for both percentiles, with
decreasing $R^I$ values for longer time aggregation periods for the covariates. For the event-scale,
both precipitations and soil moisture are outperforming other covariates, including time. The
same results are found for the annual time scale, yet with a different interpretation because annual
precipitation is representing the average level of soil moisture storage rather than event rainfall.
The link observed between the 95$^{th}$ and 99$^{th}$ percentiles with annual precipitation or soil moisture
is an indication that the long-term decrease observed for these two variables (figure 2) could be
the cause of the observed decrease in the frequency of floods above these two percentiles. At the
monthly time scale, the cumulative precipitation plays the most important role when the effects of
soil moisture, actual evapotranspiration and temperature are similar. For almost all covariates,
there is an improvement by comparison to the quantile regression model using time only.

Overall, the $R^I$ coefficients are decreasing with increasing slopes and basin mean attitude.
However, these two variables are correlated (r=0.61). This is an indication that antecedent soil
moisture condition may have a lower influence on flood generation in mountainous areas,
probably due to shallower soils and steeper slopes. For event based soil moisture and
precipitation, there is an inverse relationship with basin size: for small basins (less than 500km²)
event soil moisture and precipitation are very good predictors for the time variations of the 95$^{th}$
and the 99$^{th}$ percentiles, with $R^I$ values up to 0.6, when for larger basins the $R^I$ values are much
lower (about 0.1 to 0.2). When averaged at the monthly or annual time step, the relation is
opposite with a larger influence of soil moisture and antecedent precipitation for larger basins
with higher $R^I$ coefficients. This finding is fully consistent with results obtained for different
regions of the globe (Zhang et al., 2011, Ivancic and Shaw, 2015, Woldemeskel and Sharma,
2016, Wasko and Sharma, 2017), highlighting the buffering effects of large basins with the
capacity to store more water than smaller basins.





## 5. CONCLUSIONS

The results obtained in the present study show that despite the increase in extreme precipitation events reported by previous studies over the same domain (Ribes et al., 2018) there is not a general increase in flood occurrence. Only for a few basins, the intensity of the most extreme floods is showing significant upward trends. On the contrary, a global tendency towards fewer annual flood occurrences is observed for events of moderate intensity, above the 95$^{th}$ percentile. The same signal, with a lower magnitude, is also seen for higher floods above the 99$^{th}$ percentile. Overall, there are much more trends detected for the annual occurrence of floods than for their intensity. It should be also emphasized that the magnitude of these trends remains moderated, with only a few events less by decade and consequently these trends are only noticeable over long time periods. The decrease in soil moisture seems to be an important driver for these detected changes, indeed in all basins an increase of temperature and evapotranspiration associated with a decrease in precipitation is leading to a reduction of soil moisture over time. For several basins, the soil moisture decrease can offset the increase in extreme precipitation and generate less frequent floods. These changes are mostly observed for larger agricultural basins, with low urbanization and karstic areas. Wasko and Sharma et al. (2017) previously noted the importance of catchment size for the influence of soil moisture on flood runoff due to higher potential of soil moisture storage. The trends detected in the present work are consistent with those found in other Mediterranean regions such as Spain (Mediero et al., 2014) and Australia (Wasko and Nathan, 2019). An important finding of the present work is that with the same large scale climatic drivers (in terms of temperature, evapotranspiration and precipitation) the flood trends in the basins can be different. This shows the importance of basins characteristics to buffer climatic variability. Indeed, even if similar patterns of changes in the 95$^{th}$ and 99$^{th}$ percentiles are found, the analysis of individual catchments is revealing spatial differences even for neighboring basins caused by different topography, soil and land cover combinations. This is a factual demonstration of the commentary of Whitfield (2012) stating that is would be very difficult, if not scientifically irrelevant, to make general statements about the plausible future evolution of flood risk.



These results showing a lack of a generalized upward trend in floods should be put into
perspective with the observed increase in the vulnerability to these episodes. Indeed many reports
such as Llasat et al (2013) indicate an increase in the number of floods inducing damages
between 1981 and 2010 in South France and North Spain, which they attribute to an increased
vulnerability and land use changes. The French Mediterranean regions are concentrating 66% of
the total cost of flood damage to private properties in France (Vinet, 2011) and the total assets
lost due to floods are rising as in many other regions (CCR, 2018, Paprotny et al., 2018). The
areas close to the Mediterranean have seen a population increase and an extension of urbanized
areas, driven in part but not solely by the increase of touristic activities (Vinet, 2011, Vinet and
De Richemond, 2017). Bouwer (2011) concluded after a review of 22 disaster loss studies that
there is no trends in flood losses, corrected for changes (increases) in population and capital at
risk, which could be attributed to anthropogenic climate change". Therefore, it can be concluded
that, at least for Southern France, as noted previously by Neppel et al. (2003) the increasing cost
of damages caused by floods is rather due to the increase in socio-economic vulnerability rather
than a climate change signal towards an increase in the severity of floods. Nonetheless, the
evolution of flood frequency and intensity is a key question for risk prevention. Flood related
mortality in the Mediterranean basin is conditioned both by hazards drivers (rainfall intensity,
discharge…) but also by social drivers (behaviors, characteristics of buildings…) as shown in
different studies (Ruin et al., 2008, Vinet, 2011, Boudou et al., 2016). Deeper knowledge in
rainfall and flood trends must be crossed with exposure (e.g. population in flood prone zones) and
vulnerability data (e. g. eldering of population in the future) to anticipate evolution in human
mortality in relation with flash floods in the Mediterranean basin (Petrucci et al. 2017). As
pointed out in previous research projects (Merz et al., 2014, Meyer et al., 2014) there is a need to
integrate climate change scenarios with socio-economic change scenarios to better quantify
changes in flood risk. To achieve this task, it is necessary to develop databases on vulnerability
and exposure to be analyzed in conjunction with hydrometeorological data (Saint-Martin et al.,
491 2018).


**493 Acknowledgements**






This work is a contribution to the HYdrological cycle in The Mediterranean EXperiment (HyMeX) program, through INSU-MISTRALS support. The dataset compiled in this work are made available to the research community upon request.

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








**FIGURES**


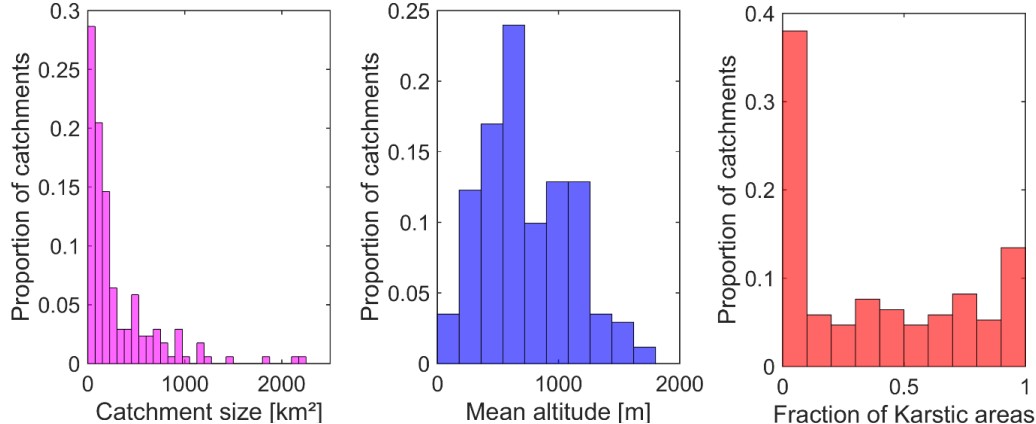

Figure 1: Catchment size, mean altitude and fraction of karstic areas



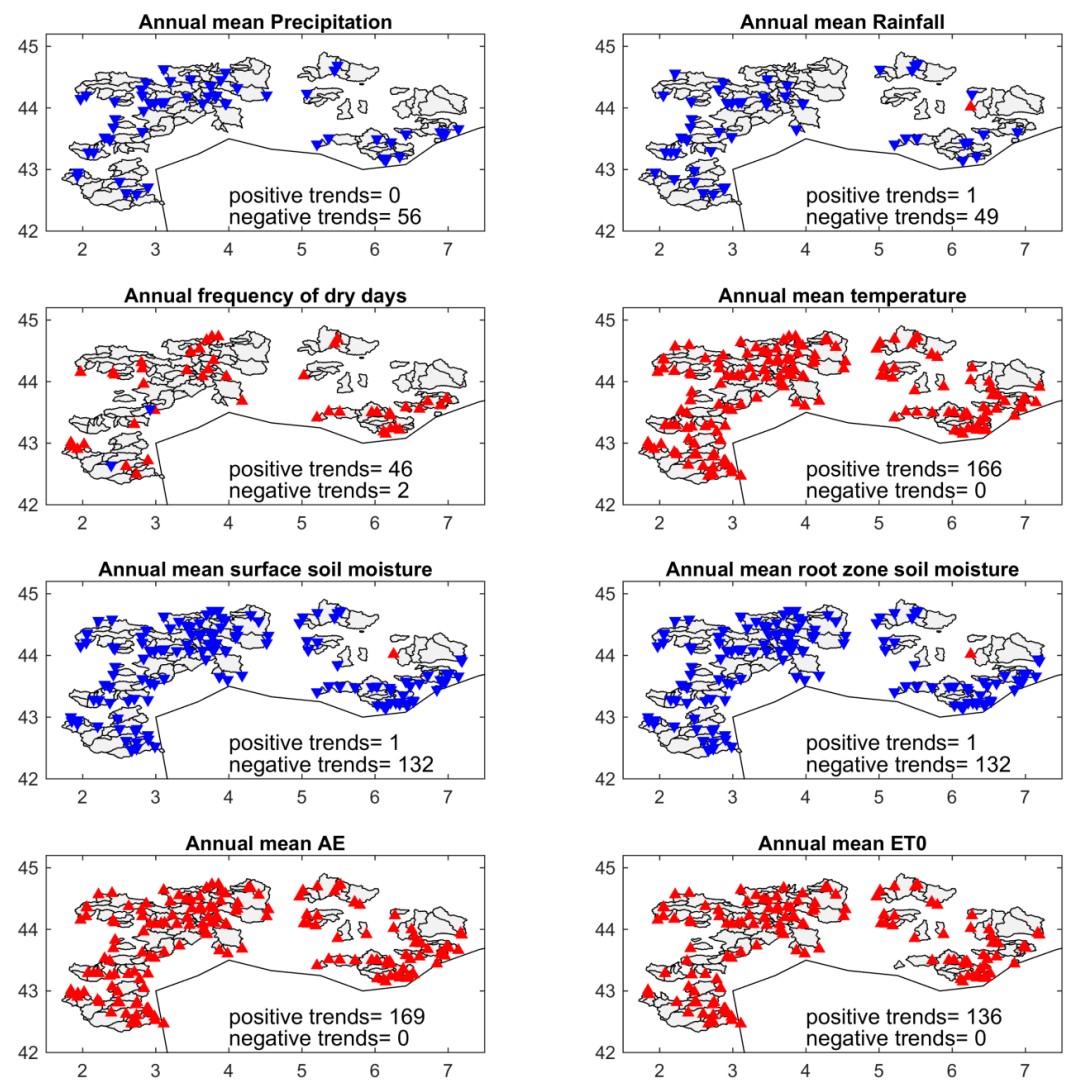


Figure 2: Annual trends between 1958 and 2018 in precipitation, temperature, soil moisture,

804        actual evapotranspiration (AE) and reference evapotranspiration (ET0).


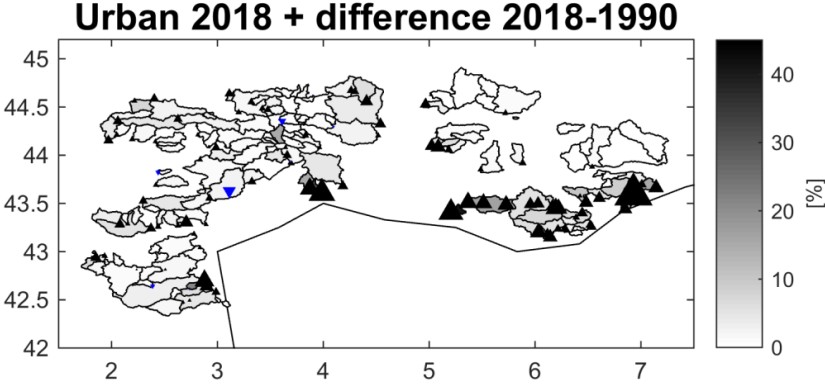

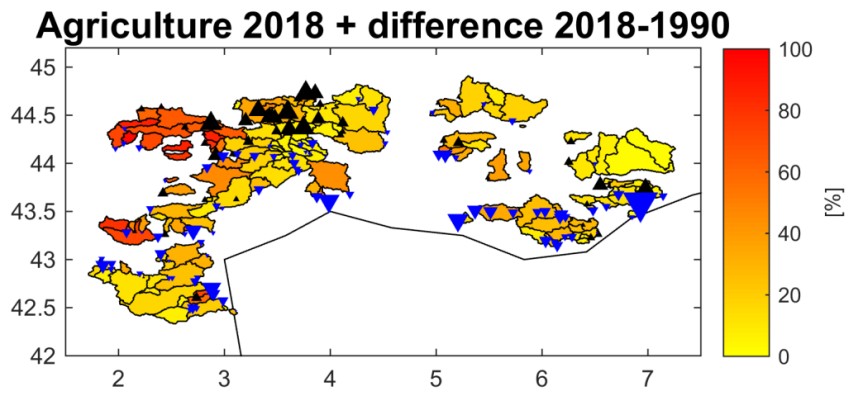

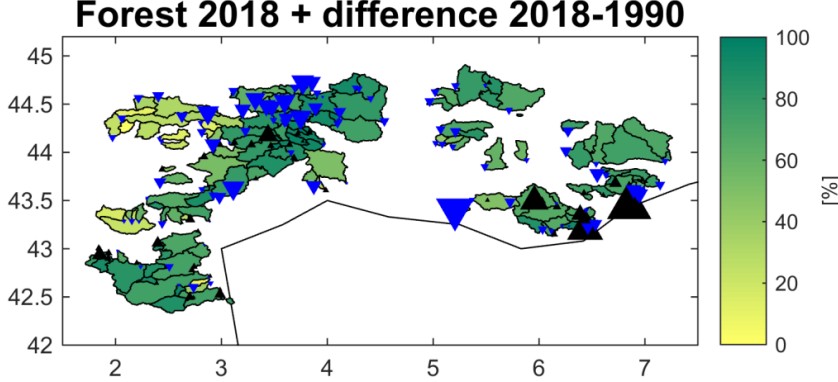


Figure 3: Urban, Agricultural and Forest cover by catchment from the CORINE database for the
year 2018 and difference between 1990 and 2018 (upward black triangles indicate an increase,
downward blue triangles a decrease).


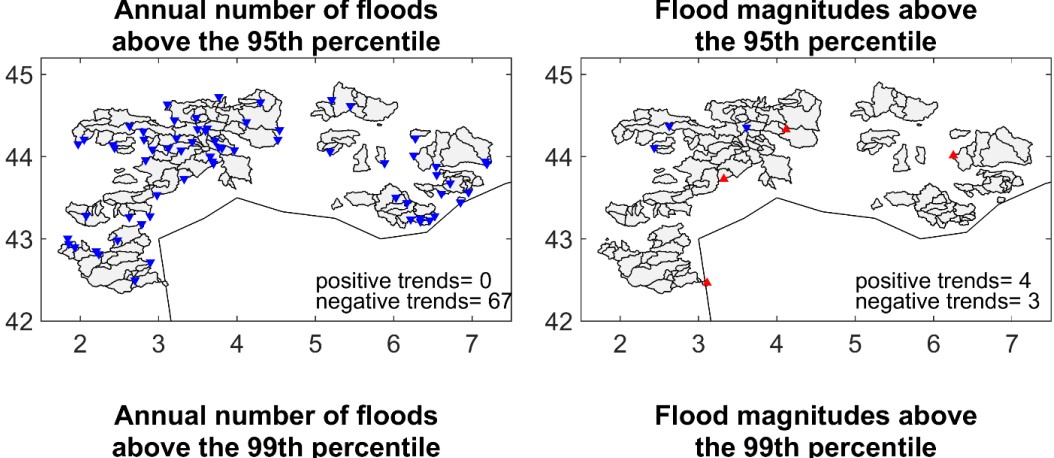

Figure 4: Trends in the annual number of flood events above the 95[th] and 99[th] percentiles (left) and in the magnitude of these threshold exceedances (right). Blue triangles indicate a decrease and red triangles an increase.














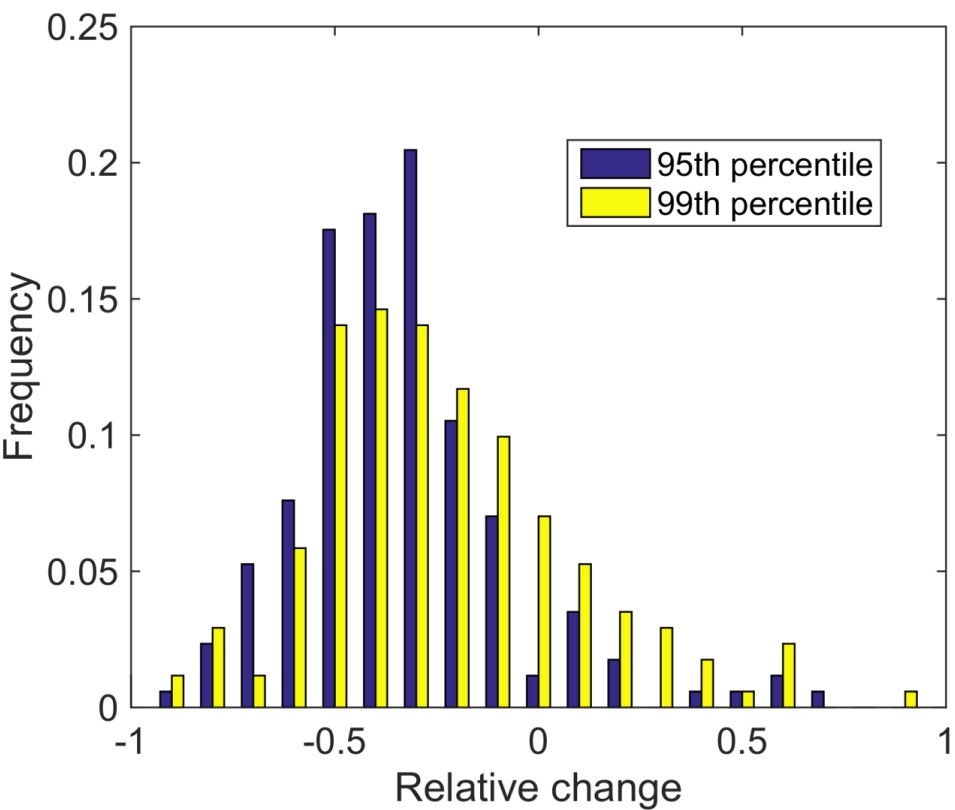



Figure 5: Histogram of the relative changes in the 95[th] and 99[th] percentile estimated from the

quantile regression models with time





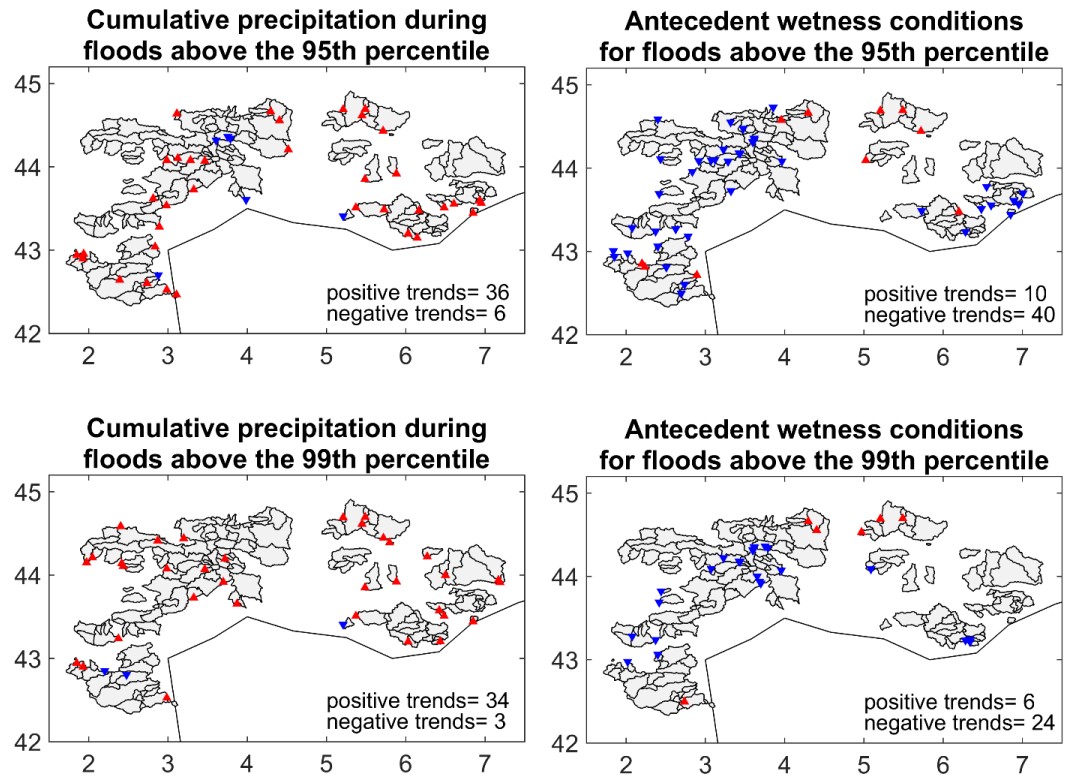


Figure 6: Trends in cumulative precipitation during flood events above the 95[th] and 99[th] percentile (left) and in the soil moisture initial conditions (right). Blue triangles indicate a decrease and red triangles an increase.










Figure 7: Distribution of the $R^I$ coefficients for different covariates for the 95$^{th}$ or 99$^{th}$ percentiles
of daily runoff, averaged at: (i) the event scale (3 days), left panels, (ii) the monthly scale, central
panels, and (ii) annual timescale, right panels.
