# Peer review of "DETECTION AND ATTRIBUTION OF FLOOD TRENDS IN MEDITERRANEAN BASSINS Tramblay, Yves1 Mimeau, Louise1 Neppel, Luc1 Vinet, Freddy2 Sauquet, Eric3 1 HSM (Univ. Montpellier, CNRS, IRD), 300 Av. du Professeur Emile Jeanbrau, 34090, Montpellier, France 2 GRED (Univ. Paul Valéry, IRD), 2 rue du Pr Henri Serres, 34000 Montpellier, France 3 IRSTEA, UR RiverLy, Centre de Lyon-Villeurbanne, 5 rue de la Doua CS"

_Hydrology and Earth System Sciences, 2019_

## Referee Comment (RC1) · Anonymous Referee #1 · 8 Jul 2019

As the authors rightly point out (lines 97-103) there is very little understanding on how changes in antecedent moisture conditions are modulating flooding under the assumption of non-stationarity. The authors not only demonstrate the modulating effect of changing antecedence on the 95th and 99th percentile of stream flow but also present results demonstrating the effect of catchment non-stationarity (e.g. urbanization). I enjoyed reading this manuscript. I find the manuscript to be novel and I recommend publication. A few minor comments are below.

General comments:

By chance of the three references I checked in the text two were missing from the bibliography e.g. Prein et al., 2016 and Bloschl et al., 2016. Please do check the referencing.

[Figure]

Line 210-217: Does quantile regression use all the data? If so, how is the 99th percentile of this comparable to the 99th percentile of the POT analysis where you only end up with a handful events per year?

Section 4.1: I get the impression that Figure 2 might not have used the Mann-Kendall test or quantile regression described in the methods but possibly a different technique? I am not sure. But this can be easily clarified but inserting a sentence at the start of this section.

Line 330: You say "was not regionally significant" does that mean that all the figures have significance only tested on a site by site basis even though you said in the methods you use a FDR? This needs to be clarified.

There a couple small grammatical things like line 48 "These results imply . . ." and Line 166: "As a very common . . ." but these are an easy fix that can be addressed at the editorial stag

Line by line:

**Line 70: A global study might make the point of high spatial variability better e.g. http://dx.doi.org/10.1029/2011GL048426; https://doi.org/10.1002/2016GL071354 But this is at the authors discretion as it may be that they were referring to variability on a smaller spatial scale (not sure because the reference was missing).**

**Line 106, 138, 482 I probably prefer "e.g" rather than ". . ." at the end of the examples. But again at authors discretion.**

**Line 120: Number of rain days or mean rainfall or both?**

**Line 217: Remove "but preliminary tests" and just write "and". This will sound more robust.**

**Line 274: Can you add the "with precipitation below 1mm" to the figure legend also please.**

**Line 325: I didn't think the trends were low? Actually they seemed quite large given the number of events per year?**

**Line 400: Is the 30 and 365 day averages also lagged or is it just the coincident month/year that is averaged?**

**Line 401: remove "rather"**

**Line 403: opposite -> "other"**

**Line 421: "mean altitude" – this typo made me chuckle!**

**Line 427: This section needs rewording I think. You say "R values up to 0.6" for small basins but to counter this you say values "about 0.1 to 0.2" for large basins. One statistic is a maximum and the other is more related to the mean. It may be larger basin values also have R values up to 0.6 but I wouldn't know? Picking a more consistent statistic would give me more confidence in these results.**

**Figures: Can it be clarified in the manuscript text and on (every) figure legend that only statistically significant sites are shown (and at what level)?**

**Figure 2: At least on this figure (but preferably on all the figures) the axes should be labelled "lat/lon" on at least one panel.**

**Figure 2 caption: Add "rainfall" to the list of variables.**

**Figure 3: Scale for triangles?**

**Figure 5: Are all sites presented here or just statistically significant ones?**

---

## Referee Comment (RC2) · Patrick Arnaud (Referee) · 10 Jul 2019

General comments:

The objective of this study is to analyze the trends on hydroclimatic data of Mediterranean catchments and research the indicators explaining these trends. The work is complete because it presents both the results of the trend tests, but it also makes the link between the different factors that can explain the significant trends. Minor revisions are proposed to improve the readability of the document and to provide some additional information that has been processed a bit quickly.

Specific comments:

L2 : replace "bassins" with "basins"

[Figure]

L69 : replace "Claussius-Clapeyron" with "Clausius-Clapeyron"

L153 ...Is the minimum duration of 20 years of data a bit limited for doing trend tests?

L174 : Can you explain the difference between "actual" et "reference evaporation".

L177 : replace "20008" with "2008"

L273 : In Figure 2, is it possible to present symbols of size proportional to the significance of the test (different threshold of p-value ), Moreover specify the difference between "Precipitation" and "Rainfall" (and there is only one in the legend of the figure).

L273 : "From figure 2, it can..."

L285 : Decrease in soil moisture for the surface and the root zone layers are due to a modelisation. It is not an observed trend. This should be clarified in the text,

L 290 : Decrease in soil moisture for the surface and the root zone layers are due to a modelisation. It is not an observed trend. This should be clarified in the text. Question: Are trends based on spatial averages on the basins have the same significance ? Is there a relationship between the calculated p-values on the trends of data averaged over each basin and the basin area? Same question for flows. One might think that the larger the basin the more the trend is "regional" and therefore less subject to sampling and therefore more significant.

L313 : a more detailed explanation would be helpful,

L319 : FDR results are analysed but not presented. In general on paragraph 4.2 : a synthetic table of values is needed or a boxplot to have an idea of Sen slope values, p-values, magnitude and number of event average.

L343 : Are the relative changes presented only for significative cases ?

L352 : these results can be related with the significativity of test on larger basin ? In

this part, no flood trends means no trend for significant test ?

L373-L387 : 34-36 basins present an increase of precipitation associated with floods. 40 basin present a decrease in antecedent soil moisture conditions. But is it the same basins. In general, in this part or in conclusion,it would be interesting to synthesize the number of basins affected by different configurations between increase, equality or decrease (significant) of rainfall, soil moisture, flood ...

L389 : replace "4.3" with "4.4"

---

## Short Comment (SC1) · 25 Jul 2019

First of all, I'd like to commend the authors for this very nice paper.

But there is one piece of information that is not clear to me, regarding the significance of the trends. The methods (3.1) explain two kinds of tests : (1) one test quantifies the significance of local trends at each station (I guess it's the MK test itself, which tells if the series is monotic, i.e. if a significant trend exists, while the method of Sen quantifies the trend) ; (2) another test, based on false discovery rates, is used to assess if the trends are significant regionally.

One question regarding the regional test: what is the corresponding spatial scale? Are all stations lumped together , or is the regional scale more "local", distinguishing for

instance Languedoc from Provence?

Besides, the articulation between these two tests and the produced figures is not clear to me. Let's take Figure 2 for instance (but the same applies to Figs 3, 4, 6): the caption does not tell if the plotted trend symbols (red and blue triangles) correspond to significant trends or not, and under which of the above tests (local, regional, or both). The text considers that the trends are significant, but provides no additional elements in this regard.

Looking more carefully at Figure 2, there is not the same number of triangles depending on the variables: is it the result of significance screening? or because there is not the same number of stations. I guess it's the the first solution, but some clarifications would enhance the paper.

Finally, I wonder if the figures with triangles to represent positive and negative trends (Figs 2,3,4,6) could not be augmented by making the size of the triangles proportional to the trend values based on Sen's slope.

It would be an interesting information with regard to the attribution targeted by the paper. And it is an interesting information per se, especially today of record-breaking heat wave in France!

Agnès Ducharne (UMR METIS, Paris)
* * *

---

## Author Comment (AC1) · 6 Sep 2019

**As the authors rightly point out (lines 97-103) there is very little understanding on how changes in antecedent moisture conditions are modulating flooding under the assumption of non-stationarity. The authors not only demonstrate the modulating effect of changing antecedence on the 95th and 99th percentile of stream flow but also present results demonstrating the effect of catchment non-stationarity (e.g. urbanization). I enjoyed reading this manuscript. I find the manuscript to be novel and I recommend publication. A few minor comments are below.**

Thank you for these positive comments about our work.

**General comments: By chance of the three references I checked in the text two were missing from the bibliography e.g. Prein et al., 2016 and Bloschl et al., 2016. Please do check the referencing.**

We have now included all references and checked the whole list.

**Line 210-217: Does quantile regression use all the data? If so, how is the 99th per-centile of this comparable to the 99th percentile of the POT analysis where you only end up with a handful events per year?**

Quantile regression only models the evolution of the $95^{th}$ or $99^{th}$ percentiles, not data above (or below). It is a convenient method to evaluate the change in different parts of the distribution and not only the mean.

Unlike the POT sampling, where all the values above the $95^{th}$ or $99^{th}$ percentiles are extracted, but then the consecutive threshold exceedances are de-clustered (i.e. only the maximum is kept) in order to avoid introducing autocorrelation.

**Section 4.1: I get the impression that Figure 2 might not have used the Mann-Kendall test or quantile regression described in the methods but possibly a different technique? I am not sure. But this can be easily clarified but inserting a sentence at the start of this section.**

We modified the caption of figure 2:
"Significant annual trends at the 10% level (Mann Kendall test) between 1958 and 2018 in precipitation, rainfall, temperature, soil moisture, actual evapotranspiration (AE) and reference evapotranspiration (ET0)."

We also clarified the text in the section as recommended:

"For each basin, the annual trends in precipitation, rainfall, temperature, soil moisture, actual and reference evapotranspiration have been analyzed with the Mann Kendall test."

**Line 330: You say "was not regionally significant" does that mean that all the figures have significance only tested on a site by site basis even though you said in the methods you use a FDR? This needs to be clarified. There a couple small grammatical things like line 48 "These results imply . . ." and Line166:**

**"As a very common . . ." but these are an easy fix that can be addressed at the editorial stag**

We tested the significance of the trends at two levels:
1- At local scale, by simply reporting for each basin if the trend for a given variable is significant, at the 10% significance level.
2- At the regional scale, since the repetition of local tests and the possible cross-correlations between the different basin's data could induce artificially more significant local values. This is why that FDR approach has been implemented, regional (or field) significance would be declared by this method if at least one local null hypothesis is rejected.

We better explain this point in the revised manuscript:

"This FDR method is applied to the MK test results to check if the trends are regionally significant. The detected trends are regionally significant if at least one local null hypothesis is rejected according to the global (or regional) significance level, $\alpha_{global}$ (Wilks, 2016). For consistency with the local trend analysis, the global significance level is also set to 10% in the FDR procedure."

**Line by line:**

**# Line 70: A global study might make the point of high spatial variability better e.g. http://dx.doi.org/10.1029/2011GL048426; https://doi.org/10.1002/2016GL071354 But this is at the authors discretion as it may be that they were referring to variability on a smaller spatial scale (not sure because the reference was missing).**

We added the reference of Wasko et al 2016 in this section.

**# Line 106, 138, 482 I probably prefer "e.g" rather than "..." at the end of the examples. But again at authors discretion.**

Changed in the text

**# Line 120: Number of rain days or mean rainfall or both?**

As written, line 120, it is the number of rain days (usually days with precip > 1 mm).

**# Line 217: Remove "but preliminary tests" and just write "and". This will sound more robust.**

Changed

**# Line 274: Can you add the "with precipitation below 1mm" to the figure legend also please**

Added

**# Line 325: I didn't think the trends were low? Actually they seemed quite large given the number of events per year?**

Indeed that is a quite subjective statement. According to the average number of events per year, the maximum trend would correspond to a about -20% decrease in the number of events. I would not say it is large, but rather moderate. We changed the text to "moderate".

**# Line 400: Is the 30 and 365 day averages also lagged or is it just the coincident month/year that is averaged?**

We agree this was not clear. We rephrased to =
At the monthly time scale representing the seasonal variability, the covariates have been averaged for the 30 days preceding the events. For the annual time scale the covariates have been averaged for 365 days preceding the events.

**# Line 401: remove "rather"**

Removed

**# Line 403: opposite -> "other"**

Changed

**# Line 421: "mean altitude" – this typo made me chuckle!**

Indeed a funny spelling error! Thanks for noticing. We replaced attitude with elevation.

**# Line 427: This section needs rewording I think. You say "R values up to 0.6" for small basins but to counter this you say values "about 0.1 to 0.2" for large basins. One statistic is a maximum and the other is more related to the mean. It may be larger basin values also have R values up to 0.6 but I wouldn't know? Picking a more consistent statistic would give me more confidence in these results.**

We modified the text as follows:
"for small basins (less than 500km²) event soil moisture and precipitation are good predictors for the time variations of the $95^{th}$ and the $99^{th}$ percentiles, with $R^1$ values up to 0.6, when for larger basins the $R^1$ values are much lower, reaching the maximum of 0.2 for some basins"

**# Figures: Can it be clarified in the manuscript text and on (every) figure legend that only statistically significant sites are shown (and at what level)?**

We modified figure captions to be more precise in each case

**# Figure 2: At least on this figure (but preferably on all the figures) the axes should be labelled "lat/lon" on at least one panel.**

We added lon/lat on all figures

**Figure 2 caption: Add "rainfall" to the list of variables.**

Added

**Figure 3: Scale for triangles?**

As requested by the other reviewers, we added on the plots the magnitude of the trends and in addition a table summarizing the trend testing results.

**Figure 5: Are all sites presented here or just statistically significant ones**

Thanks for asking this, because it helped us notice that we made a mistake by including on this figure 5 all the basins and not only those with a significant slope. We modified the figure. Now the figure is in agreement with its description in section 4.2 second paragraph.

---

## Author Comment (AC2) · 6 Sep 2019

**General comments: The objective of this study is to analyze the trends on hydroclimatic data of Mediterranean catchments and research the indicators explaining these trends. The work is complete because it presents both the results of the trend tests, but it also makes the link between the different factors that can explain the significant trends. Minor revisions are proposed to improve the readability of the document and to provide some additional information that has been processed a bit quickly.**

Thanks for this positive feedback about this work.

**Specific comments:**

**L2 : replace "bassins" with "basins"**

Changed

**L69 : replace "Claussius-Clapeyron" with "Clausius-Clapeyron"**

Changed

**L153 ...Is the minimum duration of 20 years of data a bit limited for doing trend tests?**

Yes but that is only for a few basins, when the median record length is 45 years.

**L174 : Can you explain the difference between "actual" et "reference evaporation".**

We made clearer in the text that reference evapotranspiration is computed from SAFRAN variables and actual evapotranspiration is simulated by the ISBA land surface model. Both variables are available in the SIM reanalysis over France.

Reference evapotranspiration is defined as the rate of evapotranspiration, only influenced by the atmospheric conditions, from a crop surface actively growing, completely shading the ground, well-watered, with a uniform crop height of 0.12 m, a fixed surface resistance of 70 s m−1 and an albedo of 0.23 (Allen et al.1998). In SIM, the reference evapotranspiration (ET0) is computed with the Penman-Monteith (FAO-PM) (Allen et al.1998).

Actual evapotranspiration is the quantity of water that is actually removed from the land surface due to the processes of evaporation and transpiration. It is simulated by the ISBA land surface scheme.

**L177 : replace "20008" with "2008"**

Changed

**L273 : In Figure 2, is it possible to present symbols of size proportional to the significance of the test (different threshold of p-value ), Moreover specify the**

**difference between "Precipitation" and "Rainfall" (and there is only one in the legend of the figure).**

The two other reviewers also asked to represent the magnitude of the trends in the subplots. We believe it is a bunch of information to display to have both the pvalues and the magnitude of the detected trends (the sen slope values) and would require doubling the number of figures. We choose here to add on the trend detection plots the magnitude of the detected trends.

We added precipitation (rainfall+snowfall) and rainfall in the figure caption.

**L273 : "From figure 2, it can..."L285 : Decrease in soil moisture for the surface and the root zone layers are due to a modelisation. It is not an observed trend. This should be clarified in the text,**

We agree, we added at the end of section 4.1: "Yet, it must be stressed here that the soil moisture in the present study is not observed but simulated from the ISBA land surface model"

**Question: Are trends based on spatial averages on the basins have the same significance? Is there a relationship between the calculated p-values on the trends of data averaged over each basin and the basin area? Same question for flows. One might think that the larger the basin the more the trend is "regional" and therefore less subject to sampling and therefore more significant.**

This is an interesting question, whether if we detect more trends when several SIM grid cells are averaged over large areas, in the case of large basins. We did the test for two variables, precipitation and root zone soil moisture, and we plotted for both variables the pvalues of the MK test and the Sen slope values against basin size. As shown in the figure below, there is no obvious relationship with basin size, and a quick check revealed similar results are obtained for the other variables.

[Figure]

**L313 : a more detailed explanation would be helpful,**

We added: "As noted in the method section, a declustering approach has been implemented to avoid introducing in the samples an autocorrelation signal due to several consecutive threshold exceedances belonging to the same event."

**L319 : FDR results are analysed but not presented. In general on paragraph 4.2 : a synthetic table of values is needed or a boxplot to have an idea of Sen slope values, p-values, magnitude and number of event average.**

We added a table (see below) summarizing for each variable tested the number of significant (local) trends, their sign, and the regional significance of the trends detected. In addition, as mentioned above, we added on the plots the magnitude of the trends detected. We believe these additional results give a better picture of all the analyses performed.

| | Variable | Positive trends | Negative trends | Regional significance |
|---|---|---|---|---|
| Climatic variables | Mean precipitation | 0 | 56 | Yes (28 basins) |
| | Mean rainfall | 1 | 49 | Yes (20 basins) |
| | Frequency of dry days | 46 | 2 | Yes (9 basins) |
| | Mean temperature | 166 | 0 | Yes (165 basins) |
| | Mean surface soil moisture | 1 | 132 | Yes (129 basins) |
| | Mean root zone soil moisture | 1 | 132 | Yes (129 basins) |
| | Mean actual evapotranspiration | 169 | 0 | Yes (169 basins) |
| | Mean reference evapotranspiration | 136 | 0 | Yes (131 basins) |
| Flood events | Number of floods above the 95th percentile | 0 | 67 | Yes (40 basins) |
| | Number of floods above the 99th percentile | 1 | 45 | Yes (7 basins) |
| | Flood magnitudes above the 95th percentile | 4 | 3 | No |
| | Flood magnitudes above the 99th percentile | 16 | 5 | No |
| Climatic variables associated with flood events | Cumulative precipitation during floods above the 95th percentile | 36 | 6 | Yes (16 basins) |
| | Cumulative precipitation during floods above the 99th percentile | 34 | 3 | Yes (5 basins) |
| | Antecedent wetness conditions for floods above the 95th percentile | 10 | 40 | Yes (11 basins) |
| | Antecedent wetness conditions for floods above the 95th percentile | 6 | 24 | Yes (14 basins) |

**L343 : Are the relative changes presented only for significative cases ?**

Yes, the relative changes are only presented for the significant trends. We added this information in the figure caption. We also noticed an error in the figure (see also the response to Reviewer 1) and provided a modified figure in the revised manuscript, now with the correct data plotted.

**L352 : these results can be related with the significativity of test on larger basin ? In this part, no flood trends means no trend for significant test ?**

Yes, we report only the trends that are significant at the 10% level. This is consistent throughout the paper.

As shown on the figure above, there is no evidence that trends are more often significant for larger basins for climatic drivers. For flood trends, the results are similar as shown by the figure below with the trends in the number of events above the 95[th] percentile.

[Figure]

[Figure]

**L373-L387 : 34-36 basins present an increase of precipitation associated with floods.40 basin present a decrease in antecedent soil moisture conditions. But is it the same basins. In general, in this part or in conclusion, it would be interesting to synthesize the number of basins affected by different configurations between increase, equality or decrease (significant) of rainfall, soil moisture, flood ...**

As noted below in the same paragraph, there are not necessary the same basins since "for 12 catchments an increase of event rainfall is detected when for the same catchments a decrease in the annual number of events above the 95[th] percentile is reported".

We agree about the need of a summarizing table. As noted earlier, we added a table summarizing all the results.

**L389 : replace "4.3" with "4.4"**

Changed

---

## Author Comment (AC3) · 6 Sep 2019

**First of all, I'd like to commend the authors for this very nice paper.**

Thank you

**But there is one piece of information that is not clear to me, regarding the significance of the trends. The methods (3.1) explain two kinds of tests : (1) one test quantifies the significance of local trends at each station (I guess it's the MK test itself, which tells if the series is monotic, i.e. if a significant trend exists, while the method of Sen quantifies the trend) ; (2) another test, based on false discovery rates, is used to assess if the trends are significant regionally.**

Yes, we first performed a "classical" trend analysis for each basin with the MK test. Then, we checked if the detected trends are field (or regionally) significant in a statistical sense.

**One question regarding the regional test: what is the corresponding spatial scale? Are all stations lumped together, or is the regional scale more "local", distinguishing for instance Languedoc from Provence?**

Yes all stations are lumped together. The reason why we implemented a False Discovery Rate (FDR) approach is no to provide a spatial analysis of trends for sub-regions, but to check if the multiple repetition of trend tests on different basins and variables is not introducing statistical artifacts. Basically, if you repeat a statistical test enough times, you are going to find an effect (rejection of the null hypothesis) but that effect may not actually exist.

For an excellent discussion of this aspect and about the need to implement such approaches (it is in the reference list):

Wilks, D.S.: The stippling shows statistically significant grid points: how research results are routinely overstated and over interpreted, and what to do about it, Bull. Am. Meteorol. Soc., 97, 2263–2273, 2016.

**Besides, the articulation between these two tests and the produced figures is not clear to me. Let's take Figure 2 for instance (but the same applies to Figs 3, 4, 6): the caption does not tell if the plotted trend symbols (red and blue triangles) correspond to significant trends or not, and under which of the above tests (local, regional, or both).The text considers that the trends are significant, but provides no additional elements in this regard.**

We modified the figure captions in the revised manuscript to clearly state that all the trends displayed on the plots are those significant at the 10% significance level. In addition, we provided a new table to summarize for each variable, the number of basins with significant positive trends, significant negative trends and if the detected trend signal is regionally significant.

**Looking more carefully at Figure 2, there is not the same number of triangles depending on the variables: is it the result of significance screening? or because there is not the same number of stations. I guess it's the first solution, but some clarifications would enhance the paper.**

Yes it is the first solution; significant trends for each variable are reported on these plots. This is consistent throughout the manuscript, only significant trends for each variable are reported.

**Finally, I wonder if the figures with triangles to represent positive and negative trends (Figs 2,3,4,6) could not be augmented by making the size of the triangles proportional to the trend values based on Sen's slope.**

As requested by the other reviewers, we added in the revised manuscript this information on the plots, by making the size of the triangles proportional to the trend magnitude.